# An Assessment of Oxidative Damage and Non-Enzymatic Antioxidants Status Alteration in Relation to Disease Progression in Breast Diseases

**DOI:** 10.3390/medsci4040017

**Published:** 2016-10-27

**Authors:** Kanchan Karki, Deepti Pande, Reena Negi, Ranjana S. Khanna, H.D. Khanna

**Affiliations:** 1Centre for Excellence for mountain Biology, Uttarakhand Council for Biotechnology, Haldi 263146, India; kanchan.karki1@gmail.com; 2Department of Biophysics, Institute of Medical Sciences; Banaras Hindu University, Varanasi 221005, India; deeptipande26@gmail.com (D.P.); rinkeereena2000@gmail.com (R.N.); 3Department of Chemistry, Faculty of Science, Banaras Hindu University, Varanasi 221005, India; haridevkhanna@gmail.com; 4Department of Biophysics, Institute of Medical Sciences, Banaras Hindu University, Varanasi-221005, India

**Keywords:** 8-hydroxy-2-deoxyguanosine, vitamin A, vitamin C, vitamin E, reactive oxygen species, total antioxidant status

## Abstract

The present study was aimed to evaluate the levels of oxidative stress markers in breast diseases by measuring the 8-hydroxy-2-deoxyguanosine (8-OHdG), vitamin A, vitamin C, vitamin E, and total antioxidant status (TAS) alterations in relation to cell proliferation activity and disease progression. Significant increases in the level of the oxidative damage marker 8-OHdG and cell proliferation activity were observed in breast carcinoma patients in comparison to benign and normal controls, which were accompanied by a significant decrease in non-enzymatic antioxidants and TAS concentrations (*p* < 0.05). 8-OHdG and cell proliferation levels were negatively correlated with non-enzymatic antioxidants, namely, vitamin A, vitamin C, and vitamin E levels and total antioxidant activity. Altered levels of biomarkers of oxidative stress and cell proliferation activity among the malignant, the benign, and the controls suggest a correlation of increased oxidative stress and cell proliferation activity in the progression of disease in breast carcinoma patients. In conclusion, our results showed that the characterized biomarkers (i.e., low levels of vitamin A, C and D, and the TAS status; and high levels of 8-OHdG) could be used as a suitable method for detecting subjects with malignant and benign breast diseases.

## 1. Introduction

Breast cancer affects nearly one million women per year worldwide [1]. However, with better investigation tools, a better understanding of tumor biology, and increasing public awareness, more cases are being diagnosed at earlier stages. Still, it accounts for the highest morbidity and mortality in the female population. Benign breast diseases are non-cancerous breast conditions which are at least ten times more common than breast malignancies and account for 50% of all breast biopsies performed; hence, their diagnosis, prognosis, and proper treatment is probably as important as other chronic malignancies [2]. Benign breast diseases are a heterogeneous group of lesions that are very common abnormalities of breast. Common benign lesions such as fibroadenoma, fibrocystic disease, breast abscess, ductectasia, and mastitis, as well as their relationship to the development of subsequent breast carcinogenesis, need to be more investigated.

Reactive oxygen species (ROS) and reactive nitrogen species (RNS), distinctive characteristics of cancer [3], can damage cells. If persistent, this damage may lead to base substitution, deletion, and strand fragmentation, which may inactivate tumor suppressor genes or increase the expression of proto-oncogenes within cells. Besides that, these free radicals can modify many intracellular signalling pathways via growth factor receptors and transcription factors which may modulate inflammation, angiogenesis, and cell proliferation pathways and provide a favourable environment for tumor growth [4,5,6,7,8].Previous studies have revealed the role of potential risk factors such as advancing age, early menarche, late menopause, late age at first birth, and a family history of breast cancer, as well as the imbalance in oncogenes and tumor suppressor genes in breast cancer [9,10,11] but few reports on the oxidant-antioxidant profile in the serum of patients suffering from breast carcinoma and benign breast diseases exist [12,13].

The present study investigates and provides validation of the alterations in the biochemical parameters of pro-oxidants and antioxidants using biochemical and cell-based assays in the serum of breast disease patients with respect to controls. We studied in vivo DNA damage (8-hydroxy-2-deoxyguanosine) and non-enzymatic antioxidants (i.e., vitamin A, vitamin C, vitamin E, total antioxidant capacity, and cell proliferation index) in order to determine their role in breast diseases etiology, and their application to the diagnosis of breast cancer risk groups.

## 2. Materials and Methods

### 2.1. Selection of Patients and Control Cases

For the present case control study, 60 histo-pathologically confirmed cases of benign breast disease and 60 consecutive female patients of breast carcinoma from the Department of General Surgery, University Hospital, Banaras Hindu University, Varanasi, participated. Benign breast disease patients—namely those with fibroadenoma, fibrocystic disease, breast abscess, and ductectasia—were included in the study. The patients underwent clinical breast examination, fine needle aspiration cytology, and biopsy. Patients with breast cancer were classified using the TNM Classification of Malignant Tumors system [14]. Sex matched healthy volunteers who had a socio-economic status similar to that of the patients served as controls. None of the study subjects were under oral contraceptives use, hormonal therapy, or antioxidant supplementation. Specific exclusion criteria considered for the present study were: healthy controls with acute or chronic diseases such as diabetes, parasitosis, immune dysfunction, or any other malignancy; and not being under any pharmacological therapy. The study protocol was approved by the ethical committee of the Institute of Medical Sciences, Banaras Hindu University, with the Ethical Code Dean/2012-13/177 on the date 23/3/2012. Informed consent from all participants was obtained for further analysis.

### 2.2. Analytical Methods

All biochemical and hematological investigations were done in serum collected in sterile tubes, from venous blood of patients and healthy volunteers. Collected serum was stored at −20°C for further examination.

#### 2.2.1. Reagents

All reagents used in this study were of analytical grade and obtained from Sigma Chemical Co. (St. Louis, MO, USA) and Merck (Darmstadt, Germany).

#### 2.2.2. Estimation of 8-Hydroxy-2-deoxyguanosine

Serum from the cases and control samples was used for the measurement of 8-hydroxy-2-deoxy guanosine (8-OHdG) levels using a competitive in vitro enzyme-linked immunosorbent assay (ELISA) kit obtained from Cayman Chemical Company (Ann Arbor, MI, USA) [15].

#### 2.2.3. Assay of Non-Enzymatic Antioxidants

The vitamin A level in the serum of the patients and controls was assayed by using the colorimetric method described in Paterson and Wiggins [16]. The vitamin E level was assayed using the colorimetric method described in Quaife et al. [17]; this method is based on the Emmeric–Engel reaction, which in turn is based on the reduction by tocopherols of ferric to ferrous ions that form a red complex with α,α-dipyridyl. Vitamin C level in the serum was assayed using the colorimetric method reported by McMurray and Gowenlock [18].

#### 2.2.4. Assay of Total Antioxidant Status

The total antioxidant status (TAS) in the serum was determined using Cayman’s assay kit (Ann Arbor, MI, USA) [19].

#### 2.2.5. Cell Proliferation Assay

The cell proliferation index was assayed with a Cell Proliferation ELISA, BrdU kit (Roche Diagnostics, GmbH, Penzberg, Germany), and the experiment was performed according to the manufacturer’s protocol [20].

### 2.3. Statistical Analysis

Statistical analysis was conducted with the commercial SPSS 16.0 package for Windows (SPSS, Inc., Chicago, IL, USA). All values were expressed as the mean ± standard error of the mean (SEM). Statistical significance was assessed by using Student’s *t*-test (for 2-group comparison). Categorical variables were presented as absolute numbers (frequency percentages) and analyzed by the chi-square test. All statistical analyses were 2-tailed, and a *p* value < 0.05 was considered statistically significant. Correlation between levels of oxidative stress biomarkers and cell proliferation status was evaluated with Pearson’s correlation coefficient. Odds ratios (ORs) and 95% confidence intervals (CIs) for breast cancer risk in relation to the markers of oxidative damage, non-enzymatic antioxidants, TAS, and cell proliferation index were estimated using a multinomial logistic regression analysis.

## 3. Results 

A case control study was conducted on the 60 histo-pathologically confirmed cases of breast carcinoma, the 60 cases of benign breast disease and the female healthy volunteers. This study was designed to determine the use of 8-OHdG and non-enzymatic antioxidants, namely vitamin A, vitamin E, vitamin C, total antioxidant status as biomarkers for oxidative stress, and cell proliferation activity in study subjects and controls.

### 3.1. Clinical Profile of Breast Disease Patients

Major clinical symptoms of patients suffering from breast carcinoma and benign breast diseases are depicted in Table 1. Among the study subjects, 90% (*n*=54) of benign breast disease patients had fibroadenoma, while 10% had other benign breast diseases such as fibrocystic disease (5%), duct ectasia (3.3%), and breast abscess (1.7%). Among the main symptoms of breast carcinoma and benign breast disease, all the patients showed mixed symptoms (Table 1). The masses were all unilateral and equally distributed in the left and right breast.

### 3.2. Characteristics of the Patients

Major clinical characteristics from patients and controls are depicted in Table 2. Chi-square test analysis of the studied groups showed that there were no significant differences between them in terms of age, menopausal status, parity, residence, and diet.

### 3.3. Level of Oxidative Stress Markersand Cell Proliferation Index in Benign and Malignant Breast Disease Cases and Controls

The mean ± SEM value of 8-OHdG and cell proliferation index for those with a malignant tumor was significantly higher than that for those with benign tumors and also for the control cases (Table 3). The increase in oxidative damage and cell proliferation activity of the malignant group compared with the benign and control groups was accompanied by diminished antioxidant protection. However, a similar and significant (*p* < 0.05) pattern of changes was observed in the benign group of patients as compared to their corresponding control subjects.

### 3.4. Levels of Oxidative Stress Markers and Cell Proliferation Index in Breast Carcinoma Patientsin Relation to Their Pathological Stages

Levels of oxidative damage markers, measured by 8-OHdG levels, and cell proliferation activity increased significantly with the progression of the disease while vitamin A, vitamin C, vitamin E and total antioxidant levels significantly decreased with this condition (Table 4). A significant association was observed among serum cell proliferation index and levels of oxidative stress biomarkers on the basis of clinico-pathologic stages where half of the breast carcinoma patients were in stage I/II, 50% had tumors with a diameter between 2—5 cm and an equal number of cases presented tumors with a diameter higher than 5 cm, 39 patients had clinically palpable lymph node and 10% of patients were having distant metastasis. However, serum levels of vitamin A, vitamin E and total antioxidant levels were not associated significantly with the histology of breast carcinoma where 81.67% (*n* = 49) of the carcinoma patients had ductal carcinoma while 18.33% (*n* = 11) had lobular carcinoma and medullary carcinoma.

### 3.5. Interrelationship between Cell Proliferation Activity and Oxidative Stress Markers

Correlation analysis revealed a significant association between all examined indices. A significantly negative correlation of TAS and non-enzymatic antioxidants was found between 8-OHdG and cell proliferation index. However, a significantly positive correlation was found between 8-OHdG and cell proliferation index (Table 5).

### 3.6. The Level of 8-OHdG, Non-Enzymatic Antioxidants, the Cell Proliferation Index and the Assessment of Risk of Breast Cancer

We used multinomial logistic regression analysis to identify biomarkers that can be used for classification of subjects with a risk of developing breast cancer. Table 6 shows the results of the multinomial logistic regression analyses, performed to evaluate the adjusted odds ratios of oxidative stress biomarkers and the cell proliferation index in malignant, benign, and control groups. Among the oxidative stress markers and cell proliferation index, a decrease in the non-enzymatic profile, TAS, and increased levels of 8-OHdG and the cell proliferation index emerged as the best predictive biomarkers for subjects with malignant and benign breast diseases. Similar results were observed for subjects with malignancy and controls.

## 4. Discussion

Reactive oxygen or reactive nitrogen species can damage DNA in many ways: (1) they can form a single or double-strand break, and (2) they can modify nitrogenous bases and induce cross links.

When cells are not able to rectify these damages, they undergo necrosis, show replication errors, or develop increased cell proliferation, angiogenesis, and genomic instability, which ultimately results in the onset of a variety of diseases, including breast cancer [21,22,23,24]. The most abundant by-product produced by these consequences is 8-hydroxy-2-deoxyguanosine; therefore, measurement of its levels may be applied to evaluate the load of oxidative DNA damage in the cell. This value could be important in understanding the role of oxidative stress in breast cancer development and disease intervention. In the present study, we observed significantly higher 8-OHdG serum levels in the patients suffering from breast carcinoma in comparison with benign breast disease patients and the control group. A few studies conducted by our research group have also reported an elevated level of this oxidatively modified biomolecule in breast carcinogenesis and benign breast diseases, which might be involved in the initiation of breast carcinogenesis [24,25]. Our study also indicates that the increased level of 8-OHdG in breast cancer patients was associated with disease progression and advancing to higher stages of breast carcinogenesis [25,26]. Its significantly higher level was also estimated in the benign breast diseases group in comparison to the control group.

Free radical generation and the damage caused by it in the cell is usually controlled by a large number of antioxidant systems that initiate a protection mechanism against free radicals. In the present study, a significant reduction in total antioxidant status and in non-enzymatic antioxidant profile in the malignant group in suggests and increased utilization of serum antioxidants in response to an enhanced level of oxidative damage production in those patients. The group of patients with benign breast diseases also followed a very similar significant (*p* < 0.05) pattern of alteration of antioxidant levels in comparison to healthy control subjects. Total antioxidant activity is a measure of the scavenging capacity of the cell defense system. Depleted levels of enzymatic and non-enzymatic antioxidant protective mechanisms have also been documented in a wide variety of malignancies including breast malignancy [27,28,29,30,31,32]. However, very few evidences regarding the depleted level of enzymatic and non-enzymatic antioxidant levels in fibroadenoma and other benign lesions have been available until now. Owing to the importance of antioxidants in our body, a large variety of testing methods have been proposed and applied. The colorimetric methods used by us have been validated and tested by others to quantify antioxidants levels. The relevance, advantages, and limitations of these methods have been critically discussed with respect to their chemistry and the mechanisms of antioxidant activity [33]. Apart from all these methods including the gold standard methods, the results of the present study are reproducible, and a paper has been published with regards to the estimation of antioxidants [33].

Vitamin A, a biomolecule well known for its natural antioxidant properties, plays an important role in the cellular function, development, and maintenance of normal visual acuity [34]. It has been demonstrated that it acts together with vitamin C and vitamin E to protect cells against oxidative damage [30]. Vitamin C, or ascorbic acid, is a water-soluble chain-breaking antioxidant [32] that strongly inhibits lipid peroxidation, the oxidation of glutathione, and other enzymes [35,36,37] by directly reacting with free radicals such as superoxide, hydroxyl radicals, and singlet oxygen. This interaction promotes recycling of α-tocopherol radicals and regeneration of α-tocopherol [38]. Vitamin E is the major lipid-soluble antioxidant present in lipid membranes and human plasma lipoproteins [39]. It exists in eight different isoforms, of which α-tocopherol is the most biologically active form. The main function of vitamin E is to inhibit apoptosis and to stabilize biological membranes [40]. Alpha-tocopherol also functions in vivo as a strong protector, mainly against lipid peroxidation, and prevents nitrosamine formation [38,41]. In our study, a strong negative correlation was found between an increased level of oxidative stress and a depleted level of non-enzymatic antioxidants, which results in redox imbalance and may be associated with the advancement of breast diseases.

Proliferation rates can also increase our understanding about diagnosis, prognosis, and recurrence of almost all cancers and can be used to guide treatment strategies in clinical practice. Carcinogenesis is associated with various epigenetic mechanisms which can ultimately alter cellular communication and gene expression controlling cell proliferation, differentiation, and apoptosis. An increased proliferation correlates strongly with poor prognosis. In our study, the development and continued growth of breast cancer involves significantly altered rates of cell proliferation index. 8-OHdG has been suggested as a modulator of signalling pathways related to cell proliferation and apoptosis that lead to breast cancer initiation and progression [42]. Yano et al. also found that vitamin E can suppress lung tumorigenesis by inhibiting cell proliferation at the initial stages of the disease [43]. Phenobarbital (PB) activates the transcription factor nuclear factor kappa beta (NF-κB), and dietary vitamin E effectively inhibits PB-induced NF-κB DNA binding and leads to a decreased level of cell proliferation [44]. Vitamin C can also inhibit the proliferation of A549 cells (adenocarcinomic human alveolar basal epithelial cells) in G0/G1 and S phases [45]. However, the observations with vitamin A intake are not unanimous. Small doses of vitamin A or β-carotene are suggested to prevent cancer, but in high doses this biomolecule has toxic effects [46]. The anti-cancer effects of β-carotene are such that it can elicit an anti-proliferative effect through the retinoid acid receptor (RAR) and retinoid X receptor (RXR), which can, in turn, regulate retinoid-mediated gene expression and transcription, thereby hampering cell proliferation [46]. Our study also describes a significant positive correlation between the levels of 8-OHdG and cell proliferation activity. Diminished levels of antioxidants showed a significant negative correlation with cell proliferation activity, which is in agreement with previous studies where antioxidants were found to inhibit the modulation of gene expression and cell proliferation [30,47]. Maalouf et al. found that, when vitamin E and its acetate analog were applied to cells at times before and after ultraviolet B (UVB) radiation-induced DNA fragmentation, a significant increase in the percentage of viable cells and a concomitant decrease in the number of apoptotic cells was noted, which indicates that vitamin E and its acetate analog have the potential to modulate the cellular response to UVB partly through their action on NF-κB activation [48]. Similarly, in another experiment carried out by Duarte et al., a vitamin C derivative, ascorbic acid 2-phosphate, was used to treat contact-inhibited populations of primary human dermal fibroblasts which showed faster repair of oxidatively damaged DNA bases [49]. Bagchi et al. also assessed the protective role of vitamins C, vitamin E and grape seed proanthocyanidin extract (GSPE) against smokeless tobacco (STE)-induced oxidative stress (DNA damage) in normal human oral keratinocytes (NHOK) cells. Protection values of ~11%, 26%, 28%, and 50% were recorded following the incubation with vitamin C, vitamin E, a combination of vitamins C and E, and GSPE, respectively [50].

Multinomial logistic regression was used to study association studies. The results of this study identify the contribution of the selected biomarkers; using odds ratios and the associated confidence interval, it was found that subjects with increased levels of oxidative DNA damage and cell proliferation activity or a reduced level of antioxidant defense might be at higher risk of developing breast cancer.

Taken together, these findings reveal an alteration in the tumor microenvironment due to changes in DNA damage markers and antioxidant levels. Despite the exciting advances in the field of free radical research, such applications need to be further investigated on a larger sample and in follow-up cases. Moreover, the involvement of oxidative stress in the pathogenesis of other human disorders, including various chronic malignancies, limits its usefulness as a screening tool. One of the various challenges in this field is how oxidative stress-induced cancer-related signaling pathways can be targeted for drug development. These findings provide a scientific basis for designing a treatment modality along with antioxidants supplementation; however, it cannot be generalized due to the heterogeneity and the different body requirements of every individual.

## 5. Conclusions

These findings suggest an important role of intermittent oxidative damage in the initiation and development of malignant and benign breast disease and demonstrate the involvement of oxidant-antioxidant status in mediating this process. The present study not only establishes the role of these parameters in malignant breast disease but also provides insight into the altered level of parameter profiles in benign breast disease patients, placing them in a “high-risk” category. These determinations may also be useful in establishing the pathogenic stages of breast cancer. Therefore, these indices may be employed in designing treatments synergistically with routine breast cancer management.

## Figures and Tables

**Table 1 medsci-04-00017-t001:** Clinical profile of patients with breast carcinoma and benign breast diseases.

Clinical Symptoms	Breast Carcinoma (*n* = 60)	Benign Breast Diseases (*n* = 60)
Lump		
No	8 (13.33%)	6 (10%)
Yes	52 (86.67%)	54 (90%)
Ulceration of skin/nipple		
No	45 (75%)	46 (76.7%)
Yes	15 (25%)	14 (23.3%)
Breast pain		
No	13 (21.67%)	12 (20%)
Yes	47 (78.33%)	48 (80%)
Nipple discharge		
No	55 (91.67%)	58 (96.7%)
Yes	5 (8.33%)	2 (3.3%)
Mass in axilla		
No	56 (93.33%)	57 (95%)
Yes	4 (6.67%)	3 (5%)

**Table 2 medsci-04-00017-t002:** Patient characteristics.

Parameters	Malignant (*n* = 60)	Benign (*n* = 60)	Control (*n* = 60)	χ^2^ value	*p* value
Age					
≤45 years	37 (61.67%)	41 (68.33%)	40 (66.67%)	0.640	0.726*
>45 years	23 (38.33%)	19 (31.67%)	20 (33.33%)		
Menopausal status					
Pre-menopausal	21 (35%)	15 (25%)	18 (30%)	1.429	0.490*
Post-menopausal	39 (65%)	45 (75%)	42 (70%)		
Parity					
≤3	46 (76.67%)	48 (80%)	48 (80%)	0.265	0.875*
>3	14 (23.33%)	12 (20%)	12 (20%)		
Residence					
Urban	10 (16.67%)	10 (16.67%)	11 (18.33%)	0.078	0.962*
Rural	50 (83.33%)	50 (83.33%)	49 (81.67%)		
Diet					
Vegetarian	52 (86.67%)	48 (80%)	51 (85%)	1.069	0.586*
Non-vegetarian	8 (13.33%)	12 (20%)	9 (15%)		

Categorical variables are presented as absolute numbers (frequency percentages) and were analyzed by the chi-square test. **p* value < 0.05 was considered statistically significant.

**Table 3 medsci-04-00017-t003:** Levels of non-enzymatic antioxidants, namely vitamin A, vitamin C, and vitamin E in patients with breast carcinoma, benign breast diseases, and controls.

Parameter	Serum Value			*p* value		
	Malignant	Benign	Control	Malignant/Benign	Malignant/Control	Benign/Controls
8-OHdG (pg/mL)	432.1 ± 15.6	242.2 ± 5.9	222.9 ± 6.4	*p* < 0.001*	*p* < 0.001*	*p* = 0.037*
Vitamin A (µg/dL)	55.5 ± 2.9	77.69 ± 6.8	99.7 ± 8.6	*p* = 0.003*	*p* < 0.001*	*p* = 0.047*
Vitamin C (mg/dL)	1.7 ± 0.1	3.2 ± 0.7	5.4 ± 0.7	*p* = 0.022*	*p* < 0.001*	*p* = 0.032*
Vitamin E (mg/L)	10.9 ± 0.6	15.3 ± 0.6	17.8 ± 0.9	*p* < 0.001*	*p* < 0.001*	*p* = 0.031*
Cell proliferation index (ng/mL)	1.4 ± 0.1	0.9 ± 0.1	0.7 ± 0.0	*p* < 0.001*	*p* < 0.001*	*p* = 0.006*

Data are presented as mean ± SEM. Statistical analysis was done by independent Student *t*-test. **p* value < 0.05 was considered statistically significant. 8-OHdG: 8-hydroxy-2-deoxy guanosine.

**Table 4 medsci-04-00017-t004:** Levels of non-enzymatic antioxidant in patients with breast carcinoma in relation to their clinico-pathological stage.

	8-OHdG (pg/mL)	Vitamin A (µg/dL)	Vitamin C (mg/dL)	Vitamin E (mg/L)	TAS (mmol/L)	Cell Proliferation (ng/mL)
**Stages**						
**I/II (*n* = 30)**	339.9 ± 7.8	66.2 ± 4.0	2.0 ± 0.1	13.6 ± 0.7	0.2 ± 0.0	1.0 ± 0.0
**III/IV (*n* = 30)**	511.9 ± 19.5	47.7 ± 3.4	1.3 ± 0.1	7.8 ± 0.7	0.1 ± 0.0	1.8 ± 0.1
***p* value**	<0.001*	<0.001*	<0.001*	<0.001*	<0.001*	<0.001*
**Tumor Extension**						
**1–2 (*n* = 30)**	346.1 ± 8.5	65.8 ± 4.1	1.9 ± 0.1	12.9 ± 0.8	0.2 ± 0.0	1.1 ± 0.1
**3–4 (*n* = 30)**	513.6 ± 21.0	43.0 ± 4.1	1.300 ± 0.093	8.6 ± 0.7	0.1 ± 0.0	1.7 ± 0.1
***p* value**	<0.001*	<0.001*	<0.001*	<0.001*	<0.001*	<0.001*
**Lymph Node Metastasis**						
**0 (*n* = 39)**	352.7 ± 15.3	69.1 ± 4.7	2.1 ± 0.1	13.2 ± 0.8	0.2 ± 0.0	1.0 ± 0.1
**1 (*n* = 21)**	469.3 ± 20.1	50.4 ± 3.1	1.3 ± 0.1	9.6 ± 0.8	0.1 ± 0.0	1.6 ± 0.1
***p* value**	<0.001*	0.001*	<0.001*	0.003*	0.010*	<0.001*
**Distant Metastasis**						
**0 (*n* = 54)**	402.8 ± 13.5	59.2 ± 3.0	1.7 ± 0.1	11.3 ± 0.6	0.2 ± 0.0	1.3 ± 0.1
**1 (*n* = 6)**	634.5 ± 21.6	36.3 ± 2.1	0.8 ± 0.1	4.8 ± 0.7	0.1 ± 0.0	2.2 ± 0.2
***p* value**	<0.001*	0.015*	<0.001*	0.001*	0.001*	<0.001*
**Histology**						
**Ductal (*n* = 49)**	447.1 ± 18.0	63.7 ± 6.9	2.1 ± 0.1	13.2 ± 1.1	0.2 ± 0.0	1.4 ± 0.1
**Others (*n* = 11)**	353.5 ± 15.2	55.6 ± 3.1	1.5 ± 0.1	10.3 ± 0.7	0.2 ± 0.1	1.1 ± 0.1
***p* value**	0.019*	0.278	0.006*	0.071	0.067	0.020*

Data are presented as mean ± SEM. Statistical analysis was done by independent Student *t*-test. **p* value < 0.05 was considered statistically significant. TAS: total antioxidant status.

**Table 5 medsci-04-00017-t005:** Interrelationship between cell proliferation activity and oxidative stress markers.

	Cell Proliferation Index	8-OHdG
8-OHdG	*r* = 0.53 (*p* < 0.001*)	1
Vitamin A	*r* = −0.52 (*p* < 0.001*)	*r* = −0.25 (*p* = 0.055)
Vitamin C	*r* = −0.37 (*p* = 0.019*)	*r* = −0.63 (*p* < 0.001*)
Vitamin E	*r* = −0.56 (*p* < 0.001*)	*r* = −0.60 (*p* < 0.001*)
TAS	*r* = −0.73 (*p* < 0.001*)	*r* = −0.54 (*p* < 0.001*)

**p* value < 0.05 was considered statistically significant.

**Table 6 medsci-04-00017-t006:** Multinomial regression analysis of cell proliferation and oxidative stress markers in relation to breast carcinoma risk.

	With Malignancy and Without Any Breast Disease		With Malignancy and Benign Breast Disease	
	OR (95% CI)	*p* value	OR (95% CI)	*p* value
8-OHdG	2.22 (3.33–14.82)	<0.001*	5.73 (2.34–9.68)	<0.001*
Vitamin A	0.98 (0.98–0.99)	0.025*	0.98 (0.97–0.99)	<0.001*
Vitamin C	0.79 (0.67–0.92)	0.034*	0.72 (0.62–0.84)	<0.000*
Vitamin E	0.77 (0.69–0.86)	<0.001*	0.72 (0.65–0.81)	<0.001*
TAS	1.89 (1.79–1.41)	<0.001*	1.62 (1.57–1.99)	<0.001*
Cell proliferation index	2.78 (2.60–7.13)	<0.001*	9.96 (3.45–11.54)	<0.001*

**p* value <0.05 was considered statistically significant. Odds ratio (OR) and 95% confidence interval (CI) for oxidative stress markers and cell proliferation index for subjects.

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
