# Peer review of "An Assessment of Oxidative Damage and Non-Enzymatic Antioxidants Status Alteration in Relation to Disease Progression in Breast Diseases"

_medsci, 2016, doi:10.3390/medsci4040017_

Reviewer 1 Report

The article “Assessment of in vivo oxidative stress biomarkers in 
relation to disease progression and cell proliferation 
in breast diseases” proposed by Karki et al highlight important aspects of oxidative stress in breast cancer. In spite of that, some points need to be most clear before to accept this manuscript for publish.

1.    Title – is not good and neither reflects the relevance of study. To include the design study is an option.

2.    Abstract – the differences between groups must be showed, i. e. insert values. These aspects reinforce the quality of study.

3.    Introduction – Epidemiological data from incidence, prevalence and mortality of breast cancer in India in comparison with worldwide data is relevant. The strength of study could be inserted reinforcing the relevance of this study.

4.    Methods – Please, review the inclusion/exclusion criteria……sex-matched……under pharmacological treatment?

For antioxidant analysis the gold standard methods are based in HPLC, however, in this study all vitamins were analyzed by colorimetric methods. In Discussion this aspect must be commented. Do these points represent a limitation of study? Are these methods validated and they are good concordance with gold standard methods?  

5.    Statistical analysis – the criteria used to test Odds ratio must be showed. Some Odds present values so higher. How authors can explain Odds of 150 or 199?

6.    Results -  Please, check….sex-matched.

Clinical symptoms……could be changed for Clinical profile because data showed include more information than symptoms.

Define an appropriate name for groups. The authors use in different tables……Malignant, Benign and Control AND I, II, III groups AND A, B groups.

 For Descriptive and Biochemical data use only one decimal. Colorimetric methods did not show sensitivity able to assume two or more units.

In Table 2, what is the meaning “*”?

What is the relevance to show residence data?

Why authors evaluating standard diet? This data could be commented in Discussion. Currently, high intake of meat had being associated to high oxidative stress and incidence of breast cancer. Is vegetarianism a protection factor?  Please, check these articles below:

Ingested Nitrate and Breast Cancer in the Spanish Multicase-Control Study on Cancer (MCC-Spain). Espejo-Herrera N, Gracia-Lavedan E, Pollan M, Aragonés N, Boldo E, Perez-Gomez B, Altzibar JM, Amiano P, Zabala AJ, Ardanaz E, Guevara M, Molina AJ, Barrio JP, Gómez-Acebo I, Tardón A, Peiró R, Chirlaque MD, Palau M, Muñoz M, Font-Ribera L, Castaño-Vinyals G, Kogevinas M, Villanueva CM. Environ Health Perspect. 2016 Jul;124(7):1042-9.

Red meat, poultry, and fish intake and breast cancer risk among Hispanic and Non-Hispanic white women: The Breast Cancer Health Disparities Study. Kim AE, Lundgreen A, Wolff RK, Fejerman L, John EM, Torres-MejĂ­a G, Ingles SA, Boone SD, Connor AE, Hines LM, Baumgartner KB, Giuliano A, Joshi AD, Slattery ML, Stern MC. Cancer Causes Control. 2016 Apr;27(4):527-43.

Diet and risk of breast cancer. Kotepui M. Contemp Oncol (Pozn). 2016;20(1):13-9.

Change trace elements to vitamin antioxidants according other parts of the article

The authors described MDA. Is this analysis performed? If, it were done, please insert methods and results.

Table 6 must be improved. Some values are incredible high and is not clear if author used a cutoff point to biochemical parameters.

7.    Discussion – Avoid using multiple auto citation.

Avoid discuss your results using name of groups. These names not help to understanding the real contribution of study.

Potential mechanisms associated to oxidative stress in breast cancer and benign tumor must be discussed and the relevance of design study (three groups) must be highlight.

Strength of results obtained must be compared with previous articles.

Potential limitations must be inserted.

8.    Conclusion – It must reflect a real connection between title, aims and results.

Author Response

1.      Title â€“ is not good and neither reflects the relevance of study. To include the design study is an option. Revised

2.      Abstract â€“ the differences between groups must be showed, i. e. insert values. These aspects reinforce the quality of study.

There were a lot of markers and groups therefore a significant value (p<0.05) is added

3.      Introduction â€“ Epidemiological data from incidence, prevalence and mortality of breast cancer in India in comparison with worldwide data is relevant. The strength of study could be inserted reinforcing the relevance of this study

Introduction is modified

4.      Methods â€“ Please, review the inclusion/exclusion criteria……sex-matched……under pharmacological treatment?

 Reviewed and revised

For antioxidant analysis the gold standard methods are based in HPLC, however, in this study all vitamins were analyzed by colorimetric methods. In Discussion this aspect must be commented. Do these points represent a limitation of study? Are these methods validated and they are good concordance with gold standard methods?  

These methods are validated and tested by many researchers to quantify antioxidants levels. The results are reproducible and few paper has been published with regards to the estimation of antioxidants in reputed journals and authors like to suggest few references in this regard.

OTAKI A. T, DALY J. R., AND MORTON-GILL A. Observations on oral betamethasone-17-valerate in the treatment of idiopathic steatorrhoea Gut, 1967, 8, 458

Pande D et al. NF-ÎşB p65 Subunit DNA-Binding Activity: Association with Depleted Antioxidant Levels in Breast Carcinoma Patients

Patil, N., Chavan, V.,  Karnik, N.D. 45-51 ANTIOXIDANT STATUS IN PATIENTS WITH ACUTE MYOCARDIAL INFARCTION, Indian Journal of Clinical Biochemistry, 2007;22 (1) 

5.      Statistical analysis â€“ the criteria used to test Odds ratio must be showed. Some Odds present values so higher. How authors can explain Odds of 150 or 199?

 It was written by mistake. Revised and rewritten

 6.      Results -  Please, check….sex-matched.

Sex matched is written for female patients only

Clinical symptoms……could be changed for Clinical profile because data showed include more information than symptoms.

Corrected

Define an appropriate name for groups. The authors use in different tables……Malignant, Benign and Control AND I, II, III groups AND A, B groups. Revised

For Descriptive and Biochemical data use only one decimal. Colorimetric methods did not show sensitivity able to assume two or more units. Corrected and rewritten

In Table 2, what is the meaning “*”? Corrected

What is the relevance to show residence data?

Why authors evaluating standard diet? This data could be commented in Discussion. Currently, high intake of meat had being associated to high oxidative stress and incidence of breast cancer. Is vegetarianism a protection factor?  Please, check these articles below:

Ingested Nitrate and Breast Cancer in the Spanish Multicase-Control Study on Cancer (MCC-Spain). Espejo-Herrera N, Gracia-Lavedan E, Pollan M, Aragonés N, Boldo E, Perez-Gomez B, Altzibar JM, Amiano P, Zabala AJ, Ardanaz E, Guevara M, Molina AJ, Barrio JP, Gómez-Acebo I, Tardón A, Peiró R, Chirlaque MD, Palau M, Muñoz M, Font-Ribera L, Castaño-Vinyals G, Kogevinas M, Villanueva CM. Environ Health Perspect. 2016 Jul;124(7):1042-9.

Red meat, poultry, and fish intake and breast cancer risk among Hispanic and Non-Hispanic white women: The Breast Cancer Health Disparities Study. Kim AE, Lundgreen A, Wolff RK, Fejerman L, John EM, Torres-MejĂ­a G, Ingles SA, Boone SD, Connor AE, Hines LM, Baumgartner KB, Giuliano A, Joshi AD, Slattery ML, Stern MC. Cancer Causes Control. 2016 Apr;27(4):527-43.

Diet and risk of breast cancer. Kotepui M. Contemp Oncol (Pozn). 2016;20(1):13-9.

We evaluated residence and diet data so that we can exclude patients with different socioeconomic status as diet and socioeconomic status also plays an important role during cancer initiation. There was no significant difference between the studied groups in terms of these data that means we included the patients with same patient’s characteristics.

Change trace elements to vitamin antioxidants according other parts of the article

Corrected

The authors described MDA. Is this analysis performed? If, it were done, please insert methods and results.

No we did not include MDA analysis in our present study. Text is revised and corrected

Table 6 must be improved. Some values are incredible high and is not clear if author used a cutoff point to biochemical parameters.

Revised and rewritten

Discussion â€“ Avoid using multiple auto citation.

Revised

Avoid discuss your results using name of groups. These names not help to understanding the real contribution of study.

Revised and rewritten

Potential mechanisms associated to oxidative stress in breast cancer and benign tumor must be discussed and the relevance of design study (three groups) must be highlight.

Introduction portion is Revised

Strength of results obtained must be compared with previous articles.

Few more references are added in this regard

Karihtala P1, Kauppila S, Puistola U, Jukkola-Vuorinen A.Histopathology. Divergent behaviour of oxidative stress markers 8-hydroxydeoxyguanosine (8-OHdG) and 4-hydroxy-2-nonenal (HNE) in breast carcinogenesis. (2011).58(6):854-62.

Seven A1, Erbil Y, Seven R, Inci F, Gülyaşar T, Barutçu B, Candan G. Breast cancer and benign breast disease patients evaluated in relation to oxidative stress. Cancer Biochem Biophys. (1998).16(4):333-45.

Ramaswamy G1, Krishnamoorthy L Serum carotene, vitamin A, and vitamin C levels in breast cancer and cancer of the uterine cervix. Nutr Cancer. (1996).25(2):173-7.

Maalouf S, El-Sabban M, Darwiche N et al.Protective effect of vitamin E on ultraviolet B light-induced damage in keratinocytes. Mol Carcinog (2002).34:121-130.

Duarte TL1, Cooke MS, Jones GD. Gene expression profiling reveals new protective roles for vitamin C in human skin cells. Free Radic Biol Med. (2009).46(1):78-87.

Bagchi M1, Kuszynski CA, Balmoori J, Joshi SS, Stohs SJ, Bagchi D. Protective effects of antioxidants against smokeless tobacco-induced oxidative stress and modulation of Bcl-2 and p53 genes in human oral keratinocytes. Free Radic Res ( 2001)35(2):181-94.

Potential limitations must be inserted.

Inserted in the last paragraph of discussion 

7. Conclusion â€“ It must reflect a real connection between title, aims and results.

Revised

Reviewer 2 Report

This is a thoughtful study with an interesting translational component to it. I had a fe suggestions:

1) I think the * symbol should be properly used in the tables. The authors have used the * symbol and that is followed by a comment that * in front of the p value means that it is not statistically significant. Instead of that it would be preferable if the authors use the * only in front of p values which are statistically significant.

2)Can the authors speculate on the source of the oxidative stress. Is it a mitochondrial or an extra-mitochondrial source?

3) Is there any evidence on whether patients who take nutritional anti-oxidant supplements like DHEA are protected against breast cancer? If the authors conclusion is true that decrease in levels of anti-oxidants contribute to the development of breast cancer, then patients taking supplements might have a protective effect.

Author Response

This is a thoughtful study with an interesting translational component to it. I had a fe suggestions:

1) I think the * symbol should be properly used in the tables. The authors have used the * symbol and that is followed by a comment that * in front of the p value means that it is not statistically significant. Instead of that it would be preferable if the authors use the * only in front of p values which are statistically significant.

Revised and corrected

2)Can the authors speculate on the source of the oxidative stress. Is it a mitochondrial or an extra-mitochondrial source?

No. We have not done that.

3) Is there any evidence on whether patients who take nutritional anti-oxidant supplements like DHEA are protected against breast cancer? If the authors conclusion is true that decrease in levels of anti-oxidants contribute to the development of breast cancer, then patients taking supplements might have a protective effect.

There are few studies which showed a protective role of Vitamins during oxidative stress induced damages which are included in our discussion portion.

Ramaswamy G1, Krishnamoorthy L Serum carotene, vitamin A, and vitamin C levels in breast cancer and cancer of the uterine cervix. Nutr Cancer. (1996).25(2):173-7.

Maalouf S, El-Sabban M, Darwiche N et al.Protective effect of vitamin E on ultraviolet B light-induced damage in keratinocytes. Mol Carcinog (2002).34:121-130.

Duarte TL1, Cooke MS, Jones GD. Gene expression profiling reveals new protective roles for vitamin C in human skin cells. Free Radic Biol Med. (2009).46(1):78-87.

Bagchi M1, Kuszynski CA, Balmoori J, Joshi SS, Stohs SJ, Bagchi D. Protective effects of antioxidants against smokeless tobacco-induced oxidative stress and modulation of Bcl-2 and p53 genes in human oral keratinocytes. Free Radic Res ( 2001)35(2):181-94.

Reviewer 3 Report

Comments to the Authors:

 The work by Karki and colleagues (this manuscript) attempts to correlate proliferative status to DNA damage markers in breast disease progression. The authors further investigate the levels of antioxidants in an attempt to correlate this to breast disease. In amalgam, this study looks to identify oxidative-stress related biomarkers for the progression of breast disease, particularly with benign and malignant. While the literature on antioxidant status and breast disease is well documented, this study does provide some promise in an attempt to propose a biomarker for non-malignant related breast disease. However, clarity of investigated tissue/plasma and correlations are important. Moreover, more detail as to the method for measuring cell proliferation is essential for their conclusions. Below are specific points.

Specific critique:

The conclusions of this study hinge largely on the analysis of      cell proliferation. However, the method used in this study is for isolated      cells and therefore more detailed information as to the tissue sample      assessment of proliferation is essential.

In many of the statistical analyses, students t-tests are      performed on groups greater than 2. For this, one-way ANOVA is the      recommended analysis and should be performed instead with a Bonferroni correction      for multiple groups.

8-OHdG is a marker related to oxidative stress mediated DNA      damage. While DNA damage and cancer are well known, the latter of which is      associated with cellular proliferation, it is unclear how the two      correlate in this study. Specifically, Djuric and colleagues (2001, Cancer Epidemiol Biomarkers Prev), identified 5-hydroxymethyl-2′-deoxyuridine, a DNA damage      marker, was only elevated in malignant breast disease. Perhaps more      detailed information of the protocol for proliferation would help address      this critique. These points should also feature in the discussion.

Similarly, Pan et al., (2011; BMC Cancer) and Arbranches et al., (2011; Eur J Cancer Prev)      both investigate antioxidant status with breast disease progression. In these      studies, antioxidants and vitamins were found to be protective of breast      disease progression. How does this compare to the results obtained in this      study. Especially given the strong negative correlations of some vitamins.

In table 5, care is needed for the numbers as many mistakes are      evident (e.g. Vit E proliferation = 0-.556).

Minor points: 

In the first introduction paragraph, the readers would benefit      from increased information pertaining to types of benign breast diseases;      e.g. the extensive list found in section 2.1.

There are multiple spelling/type errors that require      correcting.

Author Response

We tried to reply most of the comments.

1.      The conclusions of this study hinge largely on the analysis of  cell proliferation. However, the method used in this study is for isolated  cells and therefore more detailed information as to the tissue sample assessment of proliferation is essential.

The cell proliferation index was assayed by Cell Proliferation ELISA, BrdU kit (Roche Diagnostics, GmbH, Penzberg, Germany) and experiment was performed according to the manufacturer’s protocol

2.      In many of the statistical analyses, students t-tests are      performed on groups greater than 2. For this, one-way ANOVA is the      recommended analysis and should be performed instead with a Bonferroni correction      for multiple groups.

Yes ANOVA is the recommended analysis for multiple groups but here we are taking only two groups together at once. That’s why we performed students’t test instead of ANOVA.

3.      8-OHdG is a marker related to oxidative stress mediated DNA      damage. While DNA damage and cancer are well known, the latter of which is      associated with cellular proliferation, it is unclear how the two      correlate in this study. Specifically, Djuric and colleagues (2001, Cancer Epidemiol Biomarkers Prev), identified 5-hydroxymethyl-2′-deoxyuridine, a DNA damage      marker, was only elevated in malignant breast disease. Perhaps more      detailed information of the protocol for proliferation would help address      this critique. These points should also feature in the discussion.

Similarly, Pan et al., (2011; BMC Cancer) and Arbranches et al., (2011; Eur J Cancer Prev)      both investigate antioxidant status with breast disease progression. In these      studies, antioxidants and vitamins were found to be protective of breast      disease progression. How does this compare to the results obtained in this      study. Especially given the strong negative correlations of some vitamins.

We have included few more reference to strengthen our study There are few studies which showed a protective role of Vitamins during oxidative stress induced DNA damages

Karihtala P1, Kauppila S, Puistola U, Jukkola-Vuorinen A.Histopathology. Divergent behaviour of oxidative stress markers 8-hydroxydeoxyguanosine (8-OHdG) and 4-hydroxy-2-nonenal (HNE) in breast carcinogenesis. (2011).58(6):854-62.

Seven A1, Erbil Y, Seven R, Inci F, Gülyaşar T, Barutçu B, Candan G. Breast cancer and benign breast disease patients evaluated in relation to oxidative stress. Cancer Biochem Biophys. (1998).16(4):333-45.

Ramaswamy G1, Krishnamoorthy L Serum carotene, vitamin A, and vitamin C levels in breast cancer and cancer of the uterine cervix. Nutr Cancer. (1996).25(2):173-7.

Maalouf S, El-Sabban M, Darwiche N et al.Protective effect of vitamin E on ultraviolet B light-induced damage in keratinocytes. Mol Carcinog (2002).34:121-130.

Duarte TL1, Cooke MS, Jones GD. Gene expression profiling reveals new protective roles for vitamin C in human skin cells. Free Radic Biol Med. (2009).46(1):78-87.

Bagchi M1, Kuszynski CA, Balmoori J, Joshi SS, Stohs SJ, Bagchi D. Protective effects of antioxidants against smokeless tobacco-induced oxidative stress and modulation of Bcl-2 and p53 genes in human oral keratinocytes. Free Radic Res ( 2001)35(2):181-94.

4. In table 5, care is needed for the numbers as many mistakes are evident (e.g. Vit E proliferation = 0-.556).

Corrected and rewritten

 Minor points:  

In the first introduction paragraph, the readers would benefit      from increased information pertaining to types of benign breast diseases;      e.g. the extensive list found in section 2.1.

Revised and rewritten

There are multiple spelling/type errors that require correcting.

Corrected

Round  2

Reviewer 1 Report

The article was reviewed by authors and the most suggestions were inserted in this new version. In spide of, some points need the additional review, according below:

Please, to review the “*” in logistic regression. Is the meaning of this symbol not significant?? Is not correct to use values p = 0.000. If the values is below you must use p<0.001< p="">

The limitations or comments about antioxidant methods were not included in Discussion as recommended.

For me, is not clear what meaning sex-macthed in a study about cancer breast. Was it include men and women?

I did not identify the inclusion in potential limitation of this study. I recommend that this point will be include.

Author Response

Please, to review the “*” in logistic regression. Is the meaning of this symbol not significant?? Is not correct to use values p = 0.000. If the values is below you must use p<0.001< span="">

Corrected

The limitations or comments about antioxidant methods were not included in Discussion as recommended.

Included

For me, is not clear what meaning sex-macthed in a study about cancer breast. Was it include men and women?

No. It includes only women as we have taken female patients as cases so for controls we used the word sex matched means female (matched to cases).

I did not identify the inclusion in potential limitation of this study. I recommend that this point will be include.

This text has been included

Despite the exciting advances in the field of free radical research, such applications, however, needs to be further investigated on a larger sample size and follow-up cases. Besides that involvement of oxidative stress in the pathogenesis of other human disorders including various other chronic malignancies limits its usefulness as a screening tool. One of the various challenges in this field is how oxidative stress induced cancer-related signaling pathways be targeted for drug development. These findings provide a scientific basis to design a treatment modality along with antioxidant supplementation however it cannot be generalized due to heterogeneity and different body requirements of every individual.  

Reviewer 3 Report

            The work by Karki and colleagues (this manuscript) attempts to correlate proliferative status to DNA damage markers in breast disease progression and is much improved following the first round of review. However, one concern remains pertaining to the measure of cellular proliferation and stems from inadequate methodological explanation. The Kit used is specifically for cell based assays; however tissue samples were used in this study. It is unclear how BrdU could be incorporated into the tissues to generate a result. A detailed method is essential to understand the results from this study; especially given the results in Table 5.

Author Response

The work by Karki and colleagues (this manuscript) attempts to correlate proliferative status to DNA damage markers in breast disease progression and is much improved following the first round of review. However, one concern remains pertaining to the measure of cellular proliferation and stems from inadequate methodological explanation. The Kit used is specifically for cell based assays; however tissue samples were used in this study. It is unclear how BrdU could be incorporated into the tissues to generate a result. A detailed method is essential to understand the results from this study; especially given the results in Table 5.

We have not taken tissue samples we have taken only blood plasma samples. Table 5 shows the resuts of correlation analysis. Correlation between levels of oxidative stress biomarkers and cell proliferation status was evaluated with Pearson’s correlation coefficient.p values are corrected

Round  3

Reviewer 1 Report

After all reviews, the manuscript show a significant improvement. All points suggested were accepted by authors. So I don´t have addition comments, except for "p" values. According my last review a requested that p=0.000 were changed for the correct form (p<0.001), however, all p values were changes. In case where, p=0.034, this value must remain. Change it for p<0.05 is not correct.

Reviewer 3 Report

no further comments